# Repositioning Trimebutine Maleate as a Cancer Treatment Targeting Ovarian Cancer Stem Cells

**DOI:** 10.3390/cells10040918

**Published:** 2021-04-16

**Authors:** Heejin Lee, Oh-Bin Kwon, Jae-Eon Lee, Yong-Hyun Jeon, Dong-Seok Lee, Sang-Hyun Min, Jun-Woo Kim

**Affiliations:** 1New Drug Development Center, DGMIF, 80 Chumbok-ro, Dong-gu, Daegu 41061, Korea; free7e77@knu.ac.kr (H.L.); kob325@dgmif.re.kr (O.-B.K.); 2BK21 Plus KNU Creative BioResearch Group, School of Life Sciences and Biotechnology, Kyungpook National University, Daegu 41566, Korea; 3Laboratory Animal Center, DGMIF, 80 Chumbok-ro, Dong-gu, Daegu 41061, Korea; koof12@dgmif.re.kr (J.-E.L.); jeon9014@dgmif.re.kr (Y.-H.J.)

**Keywords:** trimebutine maleate, ovarian cancer stem cells, BKCa channel, Ca^2+^ channel, wnt/β-catenin signaling

## Abstract

The overall five-year survival rate for late-stage patients of ovarian cancer is below 29% due to disease recurrence and drug resistance. Cancer stem cells (CSCs) are known as a major contributor to drug resistance and recurrence. Accordingly, therapies targeting ovarian CSCs are needed to overcome the limitations of present treatments. This study evaluated the effect of trimebutine maleate (TM) targeting ovarian CSCs, using A2780-SP cells acquired by a sphere culture of A2780 epithelial ovarian cancer cells. TM is indicated as a gastrointestinal motility modulator and is known to as a peripheral opioid receptor agonist and a blocker for various channels. The GI50 of TM was approximately 0.4 µM in A2780-SP cells but over 100 µM in A2780 cells, demonstrating CSCs specific growth inhibition. TM induced G0/G1 arrest and increased the AV^+^/PI^+^ dead cell population in the A2780-SP samples. Furthermore, TM treatment significantly reduced tumor growth in A2780-SP xenograft mice. Voltage gated calcium channels (VGCC) and calcium-activated potassium channels (BKCa) were overexpressed on ovarian CSCs and targeted by TM; inhibition of both channels reduced A2780-SP cells viability. TM reduced stemness-related protein expression; this tendency was reproduced by the simultaneous inhibition of VGCC and BKCa compared to single channel inhibition. In addition, TM suppressed the Wnt/β-catenin, Notch, and Hedgehog pathways which contribute to many CSCs characteristics. Specifically, further suppression of the Wnt/β-catenin pathway by simultaneous inhibition of BKCa and VGCC is necessary for the effective and selective action of TM. Taken together, TM is a potential therapeutic drug for preventing ovarian cancer recurrence and drug resistance.

## 1. Introduction

Ovarian cancer is the most lethal cancer of the female genital system, with an estimated 13,940 deaths occurring in the United States in 2020 [1]. While early stage ovarian cancer can be cured using surgery, the five-year survival rate for patients diagnosed at stage IV is 29% [2]. The ovarian cancer treatment market is currently dominated by platinum- and taxane-based chemotherapy regimens [3]. In cases of advanced disease, first-line platinum-based chemotherapy can result in high (70–80%) response rates, but the majority of these patients will experience disease recurrence [4]. The most recently approved PARP inhibitor, Zejula, received marketing authorization as a maintenance treatment for all recurrent platinum-sensitive patients [5,6]. However, the need for improved first-line therapies that can delay or, ideally, prevent disease progression or recurrence is still high. This is in line with the general trend of personalized medicine observed across numerous oncology indications [7]. In particular, treatment options for patients with platinum-resistant ovarian cancer remain limited, and the prognosis for this patient group is poor. Resistance to platinum-based chemotherapy is a major factor contributing to ovarian cancer patient mortality and the requirement for effective therapies to treat this population remains the biggest unmet need [8].

Stem cells are immature cells that can undergo self-renewal and can differentiate into the cells of any organ to achieve organogenesis. Cancer stem cells (CSCs), like stem cells, can regenerate and differentiate through asymmetric division, but produce tumors with an impaired ability to regulate division [9]. Recent studies have shown that cell signal pathways that play critical roles in stem cell self-renewal, such as Notch, Sonic Hedgehog (SHH), Wnt/β-catenin, PTEN, TGF-β, and Bmi-1, also regulate CSC maintenance [10,11]. Accumulating evidence suggests that CSCs reside in various solid tumors, including ovarian cancer, where they function as a subpopulation driving tumor growth and disease progression through therapy resistance, recurrence and metastasis [12,13,14]. Therefore, treatment approaches that target CSCs may enhance responsiveness to current chemotherapy regimens and reduce the risk of tumor relapse and metastasis.

In the previous study, we tried to establish a drug screening system targeting ovarian CSCs and find effective compounds through a drug reposition strategy using an FDA-approved compound library [15]. The library was first screened for compounds selective for CSCs through sphere viability and a sphere formation assay, using a high-throughput screening system, followed by cytotoxicity testing using normal fibroblast cells. Finally, we identified that trimebutine maleate (TM) effectively reduced sphere size and viability.

TM is indicated for the treatment and relief of symptoms associated with irritable bowel syndrome and postoperative paralytic ileus to accelerate the resumption of intestinal transit following abdominal surgery [16]. It does not alter normal motility; instead, it regulates abnormal intestinal activity by increasing the number of long spike bursts in constipated patients, whereas it decreases this number in patients with diarrhea predominant IBS [17,18]. TM on the spontaneous contraction of the isolated colon gradually increased the tone, in the preparations with low tone, while it reduced the tone and the amplitude of spontaneous contraction in the preparations with high tone [19].

TM possesses a moderate opiate receptor affinity that acts on the peripheral delta, mu, and kappa receptors [20,21,22]. TM is also thought to act directly on smooth muscles by inhibiting Ca^2+^ influx and release from intracellular storage sites by blocking voltage dependent Ca^2+^ currents [23,24]. Furthermore, TM decreases the amplitude of outward K^+^ currents by inhibitory actions on both Ca^2+^-independent and-dependent types of K^+^ channel [23,25]. The sodium channel-blocking activity of TM, which is probably responsible for the inhibitory effect on glutamate release, indicates a potential therapeutic effect of these compounds in pain [26]. Although the effect and mechanism of TM as an antispasmodic drug in the intestine are well known, the therapeutic effects of TM on ovarian cancer, especially on ovarian CSCs, have not been reported.

In this study, sphere-forming cells derived from A2780 epithelial ovarian cancer cells, termed A2780-SP cells, were used as ovarian CSCs to evaluate the specific effects of TM in ovarian cancer in vitro and in vivo xenograft mouse models. We found that TM was a putative anti-ovarian CSCs drug candidate that selectively inhibited ovarian CSCs proliferation by reducing stemness and inducing CSCs specific cell death. The present research suggests TM as a drug candidate to satisfy currently unmet needs in ovarian cancer treatment, demanding the prevention of drug resistance and recurrence by selectively targeting ovarian CSCs.

## 2. Materials and Methods

### 2.1. Cell Culture

Epithelial ovarian cancer cell line A2780 and A2780-SP cells were gifted by Jae Ho Kim (Pusan National University, Republic of Korea). A2780 cells were cultured in RPMI-1640 medium (Hyclone, Logan, UT, USA) supplemented with 10% FBS (Hyclone, Logan, UT, USA) and 1% penicillin/streptomycin (Hyclone, Logan, UT, USA). Cells were detached using trypsin/EDTA solution (Hyclone, Logan, UT, USA). SKOV3 cells (ATCC, Manassas, VA, USA) were cultured in McCoy’s 5A medium (Gibco, Gaithersburg, MD, USA) supplemented with 10% fetal bovine serum (Hyclone, Logan, UT, USA) and 1% penicillin/streptomycin (Hyclone, Logan, UT, USA). OVCAR3 cells were purchased from American Type Culture Collection (ATCC, Manassas, VA, USA). OVCAR3 cells were cultured in RPMI-1640 medium (Gibco, Gaithersburg, USA) supplemented with 20% fetal bovine serum (Hyclone, Logan, UT, USA), 1% penicillin/streptomycin (Hyclone, Logan, UT, USA), and 0.01 mg/ml bovine insulin (Sigma-Aldrich, St Louis, MO, USA). A2780-SP, SKOV3-SP, OVCAR3-SP, and FACS sorted (ALDH^+^ or CD133^+^ or CD117^+^) cells were cultured in complete medium (CM) composed of Neurobasal medium (NBM, Gibco, Gaithersburg, MD, USA) supplemented with B27 (Gibco, Gaithersburg, MD, USA), HEPES (Sigma-Aldrich, St Louis, MO, USA), Glutamax (Gibco, Gaithersburg, MD, USA), 2.5 µg/mL amphotericin B (Gibco, Gaithersburg, MD, USA), 10 ng/mL basic fibroblast growth factor (bFGF) (R&D system, Minneapolis, MN, USA), 20 ng/mL human epidermal growth factor (hEGF) (R&D system, Minneapolis, MN, USA) in Ultra-Low Attachment 100 mm^2^ plate (Corning, NY, USA). Complete medium was changed every 2 to 3 days. Spheres were dissociated into single cells by treatment with Accutase (Gibco, Gaithersburg, MD, USA).

### 2.2. Cell Viability Analysis

A2780-SP, SKOV3-SP, OVCAR3-SP, and FACS sorted (ALDH^+^ or CD133^+^ or CD117^+^) cells were plated in corning Ultra-Low Attachment round bottom 96-well plates at a density of 1500 viable cells per well in complete medium and centrifuged at 3000 rpm for 3 min. A2780 cells were seeded in 96-well cell culture plates (Corning, New York, NY, USA) at a density of 1500 cells/well. After 24 h, compounds were added at a selected concentration in each well. In the cases of A2780-SP, SKOV3-SP and OVCAR3-SP cells the medium containing compound was added to each well after 3 days. Sphere cell viability was assessed after 7 days, treated compound by Cell-titer Glo (Promega, Madison, WA, USA), and luciferase was detected using TECAN plate reader (Biocompare, San Francisco, CA, USA). In case of A2780, cell viability was assessed by Cell-titer Glo (Promega, Madison, WA, USA) after 3 day treated compound.

### 2.3. Cell Sorting

To isolate the cell population with an ALDH activity, an ALDEFLUOR assay kit (STEMCELL Technologies) was used according to the manufacturer. Allophycocyanin (APC) mouse anti-human CD117 and CD133 (BD Biosciences, San Diego, CA, USA) were used to isolate CD117^+^ and CD133^+^ cells. After trypsinization, cells were washed with 1 mL cold PBS by centrifuging at 500× *g* for 5 min. Cells were suspended in PBS and incubated with anti-CD117 and anti-CD133 antibodies. After incubation for 30 min on ice in the dark, cells were washed twice with PBS, and resuspend in PBS. The CD117^+^ and CD133^+^ cells were isolated by using a Flow cytometry sorter (BD FACS Aria Ⅲ, San Diego, CA, USA).

### 2.4. Sphere Cell Proliferation Assay

A2780-SP cells were plated in corning Ultra-Low Attachment flat-bottom 96-well plates at a density of 6000 viable cells per well and grown in a CSC medium. After 24 h, A2780-SP cells were treated with the compound. Sphere cells were grown for 60 h. After 60 h, we analyzed the sphere forming confluence using IncuCyte (BioTek, Winooski, VT, USA).

### 2.5. Apoptosis (AV/PI staining) and Cell Cycle (PI Staining) Analysis

A2780-SP cells were seeded in ULA 6-well plates at a density of 1 × 10^6^ cells/well and grown in CM medium. After 24 h, the A2780-SP cells were treated with TM. Sphere cells were grown for 24 h at 37 °C in a humidified atmosphere containing 5% CO_2_. After 24 h, sphere cells were harvested by centrifugation. The supernatant was decanted, and the cells were gently re-suspended in PBS. The cells were washed once with PBS. The pelleted cells were re-suspended in 0.3 mL of PBS. To fix the cells, 0.7 mL cold ethanol was gently added dropwise to the tube containing 0.3 mL of cell suspension in PBS and left on ice for 1 h. The cells were centrifuged, washed once with cold PBS, and re-centrifuged. For Annexin V (AV)/propidium iodide (PI) staining, the cell pellet was re-suspended in 0.1 mL of AV binding buffer, followed by the addition of 5 µL of FITC AV and 5 µL PI (BD Biosciences, San Diego, CA, USA). For PI staining, the cell pellet was re-suspended in 0.1 mL of PBS, followed by the addition of 2 µL of 10 mg/mL RNase A and incubated at 37 °C for 1 h. After 1 h, 5 µL of PI solution (BD Biosciences, San Diego, CA, USA) was added. In both AV/PI and PI staining, the cells were gently vortexed and incubated for 15 min at room temperature in the dark, followed by the addition of 400 µL of AV binding buffer (AV/PI staining) or cold PBS (PI staining) to each tube. Flow cytometry was performed within 1 h.

### 2.6. Quantitative RT-PCR

The total RNA from the sample was extracted using a TRIzol RNA extraction kit (Invitrogen, Carlsbad, CA, USA) according to the manufacturer’s instructions, and 2 µg of the RNA was reverse transcribed into cDNA using GoScript^TM^ cDNA synthesis system (Promega, Madison, WA, USA). The synthesized cDNA was amplified with quantitative real-time PCR using FastStart SYBR green Master (Roche Diagnostics Ltd, Lewes, UK) and Bio-Rad S1000 Thermal cycler with the indicated primers. GAPDH was used as reference gene. The results were presented relative to control using the ddCt method. The primers used in these experiments were showed in Table 1.

### 2.7. Whole Cell Patch Clamp Recording

Conventional whole cell patch clamp experiments were performed at room temperature. The A2780 and A2780-SP cells prepared in the cover glass were moved to the recording chamber and continued circulation of the external solution. The composition of the external solution for Ca^2+^ current measurements is 143 mM NaCl, 5.6 mM KCl, 10 mM CaCl_2_, 2 mM MgCl_2_, 1 nM tetrodotoxin, 10 mM tetraethylammonium, 10 mM HEPES, 5 mM glucose and pH adjusted to 7.4 with NaOH (osmolarity, 300–310 mOsm/liter). For sodium current measurement, 10mM MgCl_2_ and 1mM tetraethylammonium, instead of 10mM CaCl_2_, 1nM tetrodotoxin and 10mM tetraethylammonium, were added. The composition of internal pipette solutions for whole cell patch clamp of Ca^2+^ and sodium current was 140 mM CsCl, 2 mM MgCl_2_, 3 mM Mg-ATP, 5 mM HEPES, 1.1 mM EGTA and pH adjusted to 7.2 with CsOH (osmolarity, 290 mOsm/liter). The composition of the external solution for BKCa channel current measurements was 140 mM NaCl, 5.4 mM KCl, 1.8 mM CaCl_2_, 1 mM MgCl_2_, 10 mM HEPES, 10 mM glucose and pH adjusted to 7.3 with NaOH. The composition of internal pipette solutions for whole cell patch clamp of BKCa current was 120 mM KCl, 110 mM K-aspartate, 1 mM MgCl_2_, 10 mM HEPES, 5 mM EGTA, 0.1 mM GTP, 5 mM Na_2_-phosphocreatine, 5 mM Mg-ATP and pH adjusted to 7.2 with KOH. Each ion current was measured using Axopatch 700B, DigiData 1440A, pClamp10.4 in voltage clamp mode, and the basic membrane potential was determined at −70 mV. The Ca^2+^ current I-V curve measured the amount of current produced by increasing the membrane voltage by +10 mV, starting with −70 mV. The Na^+^ current I-V curve measured the amount of current produced by increasing the membrane voltage by +10 mV, starting with −90 mV. The BKCa current I-V curve measured the amount of current produced by increasing the membrane voltage by +10 mV, starting with −80 mV. After measuring the normal current I-V, 10 µM of TM or manidipine was treated for 5 min, and the change in the amount of current produced by the same voltage was measured. The access resistance (Ra) value of the whole cell patch clamp was used as 10–20 MΩ.

### 2.8. CSCs Culture Condition and Western Blot Analysis

A2780-SP cells were plated in corning Ultra-Low Attachment 6-well plates at a density of 4 × 10^5^ viable cells per well and grown in complete medium. Next day compound was treated at indicated concentration for 24 h. Cells treated with same condition in Neurobasal media (NBM) or complete media (CM) were used as control. Protein extraction solution (RIPA assay buffer containing phosphatase inhibitor and protease inhibitor cocktail) was used to obtain whole cell lysates. Nuclear and cytosol fractions were prepared using a Nuclear and Cytoplasmic Isolation kit (Thermo Fisher Scientific, Waltham, MA, USA). Cell lysates were separated by SDS-PAGE gel and transferred to PVDF membranes for Western blot analysis. After blocking with 5% skim milk, the membranes were initially incubated with primary antibodies in blocking buffer overnight at 4 °C, followed by HRP-conjugated secondary antibodies for 2 h, at RT. The following primary antibodies were used: anti-phospho-AKT (Ser473) (Cell Signaling Technology, Danvers, MA, USA, #3787S), anti-AKT (Cell Signaling Technology, Danvers, MA, USA, #9272S), anti-phospho-ERK (Thr202/Tyr204) (Cell Signaling Technology, Danvers, MA, USA, #9101S), anti-ERK (Cell Signaling Technology, Danvers, MA, USA, #9102S), anti-OCT3/4 (Santa Cruz Biotechnology, CA, USA, sc-8682), anti-NANOG (Cell Signaling Technology, Danvers, MA, USA, #3580S), anti-SOX2 (Cell Signaling Technology, Danvers, MA, USA, #3579S), anti-ALDH1 (Santa Cruz Biotechnology, CA, USA, sc-166362), anti-CD133 (Abcam, Cambridge, MA, USA, ab19898) anti-beta catenin (Cell Signaling Technology, Danvers, MA, USA, #8480S), anti-phospho-beta catenin (S552) (Cell Signaling Technology, Danvers, MA, USA, #4572S), anti-Histone H3 (Cell Signaling Technology, Danvers, MA, USA, #9715S) and anti-GAPDH (Santa Cruz Biotechnology, CA, USA, sc-47724). The secondary antibodies used were goat anti-mouse IgG-HRP (Bioss, Woburn, MA, USA) and goat anti-rabbit IgG-HRP (Bioss, Woburn, MA, USA). Signals were developed with enhanced chemiluminescence HRP substrate (Bio-Rad, Hercules, CA, USA) and detected using LAS-3000 mini (Fuji film). The signal intensities were calculated with ImageJ software (NIH Image, Bethesda, MD, USA).

### 2.9. Drug Sensitivity of Ovarian CSCs in a Xenograft Tumor Model

All animal studies adhered to protocols approved by the DGMIF Institutional Animal Care and Use Committee. To assess the effect of TM in ovarian CSCs in xenograft models, A2780-SP cells (1 × 10^5^ cells) were resuspended in 50 µL matrigel solution (1:1 dilution with RPMI) and injected subcutaneously into the right and left flanks of 6- to 8-week-old female BALB/c-nu/nu mice. Mice transplanted with tumor cells were then inspected biweekly for tumor appearance on the basis of visual observation and palpation. Measurement of the length (mm), width (mm), and height (mm) of the tumor masses was performed twice weekly using electronic Vernier calipers, and the tumor volumes (mm^3^) were calculated as (length × width × height)/2. When the tumor volume was 100 mm^3^ or more, vehicle or 3 mg/kg of TM was intraperitoneally injected into nude mice daily. The size of the tumor and the body weight of the mice were measured at intervals of 3 to 4 days. All mice were euthanized on the day of the end of the experiment, and, finally, the tumor size and body weight were measured.

### 2.10. Statistical Analysis

Data were expressed as mean ± standard deviation (SD) of ≥ 3 independent experiments. Statistically significant differences were determined using 1-way ANOVA with GraphPad Prism 5 (CA, USA). A *p*-value of < 0.05 was considered statistically significant.

## 3. Results

### 3.1. TM Specifically Inhibits Ovarian CSC Growth and Induces Cell Death

The effect of TM (Figure 1a) on ovarian CSCs was tested in A2780-SP cells. Compared to their parental ovarian cancer cell line (A2780 cells), A2780-SP cells exhibited CSC characteristics. In previous reports they showed higher mRNA and protein levels of stemness-associated markers, such as *ABCG2, OCT3/4, NANOG,* and *KLF4* [15,27,28]. TM exhibited dose-dependent cell growth inhibition in A2780-SP cells (Figure 1b). Half-maximal growth inhibitory concentration (GI50) of TM was approximately 0.4 µM in A2780-SP cells but over 100 µM in A2780 cells, demonstrating specific growth inhibition for ovarian CSCs. In contrast, cisplatin, which is clinically used as a primary ovarian cancer treatment, exhibited greater sensitivity to A2780 cells than A2780-SP cells (Figure 1b); cisplatin GI50 was approximately 50 µM in A2780 cells and 100 µM in A2780-SP cells. Similar results were found in other epithelial ovarian cancer cells (SKOV3 and OVCAR3). TM showed greater cell growth inhibition than cisplatin on SKOV3-SP cells (derived from SKOV3) and OVCAR3-SP cells (derived from OVCAR3), but the inhibitory degree was weaker than that in A2780-SP cells (Appendix A).

A large number of subpopulations with enhanced tumor initiating capacity have been identified, especially several markers, including CD24, CD44, CD117, CD133, ABCG, ESA and ALDH, which are widely used in the literature to identify and investigate human epithelial cancer stem cells. In particular, multiple markers, including CD44, CD117, CD133 and ALDH, are utilized to identify ovarian CSCs [14,29]. TM showed a weaker effect in cells sorted by ovarian CSC markers, such as ALDH, CD117, or CD133 from A2780 cells (Appendix A). These data suggest that ovarian CSCs are comprised of various traits, and that TM demonstrates a stronger effect on subpopulations expressing other traits or markers of ovarian CSCs.

The proliferation curve for TM in A2780-SP cells using real-time live imaging also showed that TM dose-dependently inhibited A2780-SP cell proliferation (Figure 1c). To investigate whether TM could induce cell cycle arrest and apoptosis in ovarian CSCs, we performed PI staining and an AV/PI apoptosis assay (Figure 1d and Appendix A). After 24 h of TM treatment, cell cycle stage and apoptosis were examined using flow cytometry. TM induced G0/G1 arrest and significantly increased the population of AV^+^/PI^+^ cells, meaning late stage apoptotic and dead cells in A2780-SP samples at concentrations over 1 µM. 

Next, we examined whether TM could inhibit tumor growth by ovarian CSCs under physiological environment of in vivo using A2780-SP xenograft mouse model. TM was injected intraperitoneally at a dose of 3 mg / kg daily. The result showed that TM treatment significantly reduced tumor growth in A2780-SP xenograft mice, correlated with in vitro result (Figure 1d). The lack of toxicity was demonstrated by the unchanged body weight of the test mice (Figure 1e). These results suggested that TM inhibited the in vivo tumor growth by ovarian CSCs.

### 3.2. BKCa and Sodium Channel Subunits Are Overexpressed on Ovarian CSCs

Next, we intended to identify the mechanism by which TM specifically induced ovarian CSC death. Because TM can act as an opioid receptor agonist an antispasmodic reaction, we first examined the relative expression level of the opioid receptor in A2780-SP cells. Expression levels of opioid receptor family genes, such as *OPRM1* and *OPRK1*, did not significantly differ between A2780 and A2780-SP cells (Appendix A). Opioid receptor agonists, such as DAMGO and dynorphin, did not inhibit cell growth in A2780-SP cells, indicating that the action of TM in ovarian CSCs did not come from the inhibition of opioid receptors (Appendix A). Because TM acts an antispasmodic by inhibiting multiple channels, including the voltage gated calcium channel, sodium channel, or large conductance calcium activated potassium channel (BKCa), we examined the relative expression level of these channels in A2780-SP cells. In the case of the calcium channel, we had previously showed that *CACNA1D*, *CACNA1F*, and *CACNA1H* genes were overexpressed in A2780-SP cells compared to A2780 cells [15], and that these genes were important for maintaining stemness. Here, we confirmed the relative mRNA expression levels of BKCa genes (*KCMA1*, *KCMB2*, *KCMB3*, and *KCMB4*) (Figure 2a) and sodium channel genes (*SCN5A, SCN8A, SCN1B*, and *SCN4B*) (Figure 2b) in A2780-SP cells compared to A2780 cells. KCNMA1, BKCa gene and *SCN4B* sodium channel gene were highly expressed in A2780-SP cells.

### 3.3. TM Can Inhibit Ca^2+^, Na^+^ and BKCa Channel Currents in Ovarian CSCs

We measured differences in the currents of sodium, L-type calcium, and BKCa channels between A2780 and A2780-SP cells using a whole-cell patch clamp amplifier to test whether channel gene overexpression was related to the specific action of TM on ovarian CSCs (Figure 2c–e). Each current was higher in A2780-SP cells than in A2780 cells. A 10 µM concentration of TM decreased the increased amplitude of L-type Ca^2+^ (Figure 2c), BKCa (Figure 2d), and sodium currents (Figure 2e) in A2780-SP cells, suggesting that the calcium, BKCa, and sodium channels are highly expressed or activated in A2780-SP cells and that TM can directly act on those channels in A2780-SP cells. We had previously shown that L-type calcium channel blockers (CCBs) such as manidipine reduce the amplitude of L-type Ca^2+^ currents [15]. Our present result showed that TM reduced the amplitude of L-type Ca^2+^ currents but the effect was weaker than that of manidipine.

### 3.4. Simultaneous Inhibition of BK and Calcium Channels Amplifies Cell Growth Inhibition in Ovarian CSCs

To confirm which channels were important to maintaining CSC proliferation, inhibitors of each channel were used (manidipine as a calcium channel blocker, paxilline as a BKCa channel blocker, and tetrodotoxin (TTX) as a sodium channel blocker). Paxilline did not effectively inhibit the growth of A2780-SP (GI50 of 10 µM) and showed similar activity between A2780 and A2780-SP cells. TTX did not inhibit the growth of A2780-SP or A2780 cells, even at a concentration of up to 100 µM. Manidipine inhibited A2780-SP cell growth at concentrations of 1–10uM and showed 10-fold better activity compared to that in A2780 cells (Appendix A).

Although manidipine showed the strongest L-type Ca^2+^ channel inhibition in A2780-SP cells, its growth inhibition was weaker than that of TM. In contrast, A2780 growth inhibition by manidipine was stronger than that by TM. Therefore, the independent inhibition of each channel could not explain the specific effect of TM in ovarian CSCs. So, we examined the effect of simultaneously inhibiting both channels on A2780 and A2780-SP cell growth. A2780 cell viability did not change when co-treated with two of manidipine, paxilline, and TTX (Figure 3a). In contrast, cell growth inhibition in A2780-SP cells was further increased when manidipine and paxilline were simultaneously applied (Figure 3b). The effect of simultaneously inhibiting the calcium and sodium channels or BKCa and sodium channels was weaker than the simultaneous inhibition of the calcium and BKCa channels in A2780-SP cells (Figure 3b). Simultaneous inhibition of the calcium channel and various potassium channels did not further reduce A2780-SP cell growth (Appendix A), suggesting that TM acts more potently and selectively by simultaneous inhibition of the calcium and BKCa channels in A2780-SP cells.

### 3.5. TM Further Reduces Stemness-Related Transcription Factors by the Simultaneous Inhibition of BK and Calcium Channel in Ovarian CSCs

There are a number of stemness-related transcription factors, such as Nanog (nanog homeobox), Oct4 (octamer binding transcription factor 4) and Sox2 (sex determining region Y-related high mobility group box 2), which are activated in both embryonic and cancer stem cells and comprise a core regulatory network for stem cell self-renew and pluripotency maintenance [30]. The ERK/ MAPK and PI3K/AKT pathways are two critical pathways involved in tumorigenicity, including cellular proliferation, growth, survival and mobility, and deregulated in various cancers and CSCs [31,32,33]. We examined the effect of TM on the activation of these pathways in CSCs culture condition (CM-cultured A2780-SPs). TM especially reduced OCT3/4 and SOX2 protein expression related to stemness and ERK phosphorylation related to cell growth, respectively (Figure 3c). An examination of signaling molecule reduction using manidipine and paxilline for the simultaneous inhibition of the calcium and BKCa channel showed that OCT3/4 and SOX2 protein expression was further reduced by simultaneous inhibition than it was by single inhibition, whereas ERK phosphorylation was not influenced by simultaneous inhibition (Figure 3d). These results confirmed that TM could significantly decrease the stem cell properties of A2780-SP cells through simultaneous inhibition of the calcium and BKCa channels.

### 3.6. TM Further Reduces Wnt/β-Catenin Signaling by Dual Inhibition of BK and Calcium Channels in Ovarian CSCs

Since the preceding result showed TM could decrease stemness-related transcription factors through the simultaneous blockade of calcium and BKCa channels, we next sought to find the signaling pathway linking the reduction of stemness-related transcription factor to the blockade of calcium and BKCa channels. A number of developmental pathways that control self-renewal of normal stem cells, and have important roles in embryonic development and differentiation, are also critical for the maintenance and self-renewal of CSCs, which include Wnt/β-catenin, Hedgehog (HH), and Notch signaling pathways [11,30,34]. So, we tested whether each of signal pathways regulating stemness was changed by TM in CSCs culture condition (CM-cultured A2780-SPs). mRNA level of *GLI-1, HIP1* and *PTCH-1* for HH pathway [35,36], mRNA level of *HES1* and *c-Myc* for Notch pathway [37], and a translocation of β-catenin to the nucleus and β-catenin phosphorylation on Ser 552, associated with enhanced β-catenin transcription for Wnt/β-catenin pathway [38], were used as markers for each signaling pathway. The relative mRNA levels of HH target genes (*GLI-1* and *HIP-1*) and the Notch target gene (*c-MYC*) were highly expressed in CSCs culture condition compared to Neurobasal media (NBM) cultured A2780-SPs; these mRNA levels were significantly decreased by TM (Figure 4a,b).

The translocation of β-catenin to the nucleus was increased in CSCs culture condition but decreased when treated with TM (Figure 4c). In particular, the phosphorylation on Ser 552 associated with enhanced β-catenin transcription was increased in CSCs culture condition and TM reduced β-catenin phosphorylation on Ser 552. (Figure 4d). Next, we confirmed that the inhibition of these signaling pathways was caused by the simultaneous inhibition of the calcium and BKCa channels using manidipine and paxilline. The translocation of β-catenin to the nucleus (Figure 4e) and phosphorylation on Ser 552 (Figure 4f) were further reduced by simultaneous inhibition than they were by independent inhibition. Notch and HH signaling was not influenced by simultaneous inhibition (Appendix A). This confirmed that TM could significantly decrease the stem cell properties of A2780-SP cells by further reducing Wnt/β-catenin signaling through simultaneously inhibiting the calcium and BKCa channels.

## 4. Discussion

In the current study, we have shown that TM specifically inhibits proliferation, reduces stemness, and induces cell death in ovarian CSCs. The xenograft mouse model has shown that TM can target ovarian CSCs in a physiological environment as well as in an artificial in vitro environment that maintains CSC properties. We demonstrated that further suppression of the Wnt signaling pathway by simultaneous inhibition of BKCa and Ca^2+^ channels is required for the effective and selective action of TM.

It is well known that TM can act on an antispasmodic reaction as an opioid receptor agonist [20,21,22]. According to previous reports, [39,40], a wide variety of synthetic and natural opioids, including those specific for μ (DAMGO, endomorphin-1 and -2), δ (DPDPE) and κ (U69593) opioid receptors, showed that none of these compounds had any effect on growth of ovarian cancer cell, except for opioid growth factor (OGF), endogenous opioid peptide. OGF showed the efficacy through OGF receptor in ovarian and hepatocellular cancer. Our data also showed that opioid receptor expression in ovarian CSCs was not high and that opioid receptor agonists, such as DAMGO and dynorphin, had no effect, thereby demonstrating TM did not act through the opioid receptor in ovarian CSCs. It was also reported that TM acts as an opioid receptor agonist in glioma and glioblastoma cells, and inhibits cell proliferation at a range of 50–400µM without difference between glioma and glioblastoma cells by inhibiting ERK and AKT signaling pathways [41]. On the other hand, our data showed that TM could act on ovarian CSCs more selectively and effectively by the inhibition of stemness-related transcription factors, such as OCT3/4 and SOX2, through a blockade of the calcium and BKCa channels that are overexpressed in the ovarian CSCs.

BKCa and calcium channels have been found in many and various tumor cells, including prostate cancer, breast cancer, gliomas, melanoma, colon cancer, and other tumors [42,43]. We previously showed that L- and T-type calcium channel genes, such as *CACNA1D, CACNA1F,* and *CACNA1H*, are highly expressed and related to stemness in ovarian CSCs [15]. CCBs such as manidipine significantly decreased CSC-associated gene expression in ovarian CSCs and induced apoptosis, suggesting that these calcium channel genes were important for maintaining the stem cell characteristics of ovarian CSCs. The present results show that the BKCa subunit *KCNMA1* gene is overexpressed, and that the BKCa current is amplified in A2780-SP cells as ovarian CSCs. *KCNMA1* gene amplification was correlated with, and contributed to a high proliferation rate, malignancy, invasion and metastasis to brain and breast cancer [44,45]. *KCNMA1* gene amplification was also associated with progression to late-stage, metastatic, and hormone-refractory human prostate cancer, and was associated with a high BKCa channel density in the cellular membrane in the PC-3 prostate cancer cell line [46]. Amplified BKCa channels might possess augmented sensitivity to Ca^2+^ and voltage, generating K^+^ currents in environments and shifting the cellular transmembrane potential to favor cancer cell proliferation, migration, and invasion [45,47,48,49]. Tumorigenicity attenuation in breast cancer by BKCa channel inhibition was a result of depolarizing shifts in cell transmembrane potential and subsequent downregulation of β-catenin and (phospho) Akt and HER-2/neu protein levels [50].

Here, we show that TM has less ability to inhibit calcium channels in ovarian CSCs compared to manidipine; TM is expected to be less capable of inhibiting calcium channels compared to other commonly used CCB drugs. This weak calcium channel blocking potency may explain more CSCs-selective effect of TM compared to CCB drugs. On the other hand, other factors besides calcium channel blocking potency are needed to explain the superior anti-CSCs effect of TM over CCBs. The sole inhibition of the BKCa channel by paxilline hardly inhibited cell growth in ovarian CSCs and cancer cells. It seems that the effect of BKCa inhibition can be counteracted by increasing the intracellular Ca^2+^ concentration using a voltage-gated calcium channel as positive feedback. Of note, the simultaneous inhibition of the BKCa channel and the Ca^2+^ channel overexpressed on ovarian CSCs by TM could further inhibit Wnt/β-catenin signaling and further reduce stemness-related transcription factors, thereby showing greater efficacy than inhibiting only the Ca^2+^ channel in ovarian CSCs. It was reported that translocated β-catenin to the nucleus through canonical Wnt signaling directly interacted with Tcf3 to repress pluripotency transcription factors [51], and, in gastric cancer cells, Helicobacter pylori activated Wnt/ β-catenin signaling which, in turn, upregulates Nanog and Oct4 to promote CSC-like properties [52].

The expression of BKCa-Ca^2+^ channel complexes in tumor cells has been previously reported [42]. Calcium channels act as Ca^2+^ amplifiers, provide sufficient Ca^2+^ for BKCa activity, and drive essential biological functions for tumor development in cancer cells. The activated BKCa channel further controls the membrane potential suitable for cell proliferation. A recent paper describes a role of the BKCa-Cav3.2 complex localized in the same plasma membrane areas in LNCaP prostate cancer cell proliferation by controlling a constitutive Ca^2+^ entry channel [53]. Therefore, it seems that each channel works in concert with the other channel for cell proliferation. Our results show that, when one channel is suppressed, especially in a system which both channels are overexpressed, such as ovarian CSCs, both channels act complementarily to each other to maintain cell proliferation and, thus, simultaneous inhibition of both channels is necessary for effective cell growth inhibition. 

TM shows a stronger effect on A2780-SP cells than other ovarian CSCs based on the Appendix A. There may be differences in the level of expression of BKCa and Ca channels for each subpopulation of ovarian CSCs, and TM may have the strongest potency in A2780-SP cells, which exhibit the highest expression levels of BKCa and Ca channels. The improvement of survival rate in aggressive cancers through individualized predictive, preventive, and personalized medicine requires excavating predictive biomarkers that will enable the selection of patients who would benefit from a therapy [54]. Biomarkers such as carbohydrate antigen 125 (CA125), human epididymis protein 4 (HE4), breast cancer 1 (BRCA1), and human chorionic gonadotropin (HCG) are used for the diagnosis of ovarian cancers, but the diagnosis of ovarian cancers based on these common biomarkers remains unsatisfactory [55]. BKCa and calcium channels may be a suitable pharmacological target for treating recurrence and drug resistance in late-stage ovarian cancer patients who exhibit amplification of the *KCNMA1* gene and calcium channel subunits genes (*CACNA1D*, *CACNA1F,* and *CACNA1H*). It is necessary to verify whether these genes are common CSC markers or are representative of specific subpopulations, and whether TM can improve the survival rate of patients highly expressing these genes in combination with conventional treatments. Customized patient group selection and clinical design, along with the application of this information, may increase the likelihood of success in clinical trials.

Paxilline is not an approved drug and is a toxic alkaloid. Therefore, although paxilline can be used to show the inhibitory effect of BKCa channel in vitro, it is difficult to develop it as a drug. Since manidipine also needs to be re-approved as a cancer treatment, there will be many difficulties in making a successful clinical design with combination of the two. On the other hand, the advantages of repositioning TM to target ovarian CSCs over synergistically using paxilline and manidipine are that it is superior in terms of safety, and has a higher possibility of success because it requires a clinical design considering the PK and toxicity of only one drug. Since TM is available in many countries and in various forms, extensive accumulated clinical experience has shown that it is active in many indications with a good safety profile [16]. Clinically, TM is effective in treating both acute and chronic abdominal pain in patients with functional bowel disorders, especially irritable bowel syndrome, at doses ranging from 300 to 600 mg/day. Oral TM administration in animals results in nearly complete intestinal absorption, and 94% of absorbed TM is eliminated by the kidneys as various metabolites. Peak plasma concentration is observed 1 h after ingestion. Reported side-effects include skin rash, sleepiness, and very rare cases of headache, dry mouth, constipation, diarrhea, vomiting, asthenia, and dizziness, at the same frequency as in patients treated with a placebo. We showed 3mpk IP injection of TM effectively reduced tumor size in ovarian CSCs mouse xenograft model. Changes in the dosage and administration route used for a gastrointestinal motility modulating agent demand another effort to prove a safety profile for clinical trials, but these may be a drug development strategy to prevent an alternative prescription that is used as anti-CSCs of gastrointestinal motility modulating therapy. It is necessary to confirm an effective dose and administration route to clinically target ovarian CSCs. In vitro and in vivo biological trials to further understand the disease evolution of ovarian cancer and the resistance mechanism developed by tumors, as well as to validate the action of TM, are needed for clinical success.

## 5. Conclusions

We report that TM efficiently and selectively suppresses stemness and proliferation of ovarian CSCs, according to the suggested mechanism (Figure 5). It is possible that the Wnt/ β-catenin signaling may partially contribute to the action of TM, and other signaling pathways not examined are needed to explain the exact mechanism of TM. Extensive data analysis of changes in expression patterns, such as RNA sequencing, will be helpful. The heterogeneous trait of CSCs makes the development of CSCs target therapy difficult. To overcome these obstacles, more biological efforts to characterize CSCs, to understand mechanisms of disease evolution and the resistance mechanism developed by CSCs, and to find more CSC specific drug candidates based on these mechanisms are required. The current research is part of these efforts and may provide a basis for satisfying the unmet needs of current cancer treatments.

## 6. Patents

The application number of patent related to this work is PCT/KR2020/004637.

## Figures and Tables

**Figure 1 cells-10-00918-f001:**
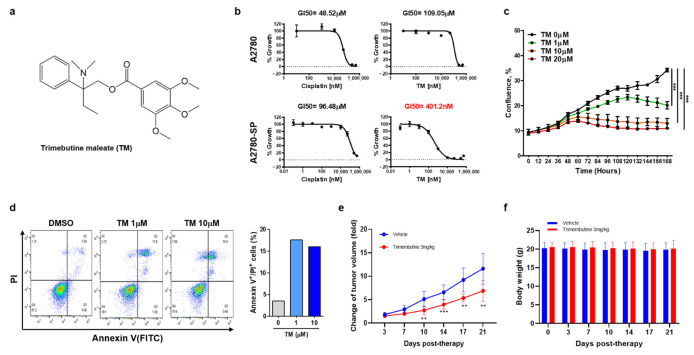
Trimebutine(TM) inhibits ovarian CSCs growth more specifically than ovarian cancer cells, and induces cell death. (**a**) Trimebutine maleate (TM) structure. (**b**) ATP-based cell viability was measured after 3 days (A2780) or 7 days (A2780-SP) of TM or cisplatin treatment at the indicated concentrations. Data are expressed as the mean ± SEM in triplicate. Nonlinear regression analysis was performed to obtain GI50 values using Prism 6. (**c**) The proliferation curve of A2780-SP cells was obtained using real-time live imaging for 60 h after TM treatment. The data are expressed as the mean ± SD of two or three independent experiments. *** *p* < 0.001. (**d**) AV/PI apoptosis assay. After 24 h treatment with either DMSO or TM, apoptotic cells were examined using flow cytometry. (**e**) The effect of tumor growth inhibition by TM in the A2780-SP xenograft model. (**f**) The effect of body weight change by TM in the A2780-SP xenograft model. The data are expressed as the mean ± SD of three independent experiments. ** *p* < 0.01, *** *p* < 0.001.

**Figure 2 cells-10-00918-f002:**
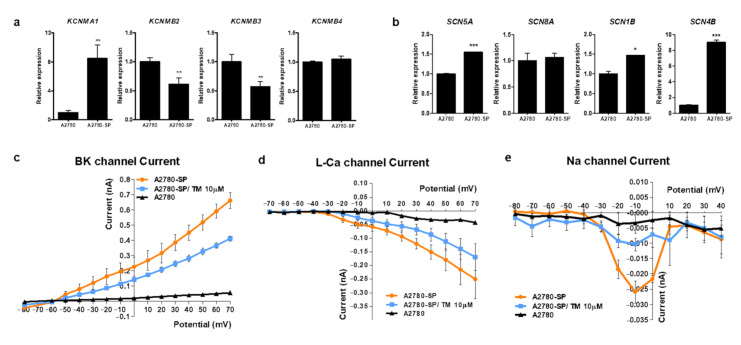
Subunits of BKCa and Na^+^ channel are overexpressed on ovarian CSCs and TM can inhibit the current of Ca^2+^, BKCa and Na^+^ channels. Compared mRNA expression levels of (**a**) BKCa channel genes (*KCMA1, KCMB2, KCMB3*, and *KCMB4*) and (**b**) voltage gated sodium channel genes (*SCN5A, SCN8A, SCN1B*, and *SCN4B*) in A2780-SP and A2780 cells, quantified by real-time PCR analysis. P values were determined by one-way ANOVA and GraphPad Prism 5 software (* *p* < 0.05; ** *p* < 0.01; *** *p* < 0.001). Currents of calcium (**c**), BKCa (**d**), and Na^+^ (**e**) in A2780 and A2780-SP cells, measured using the whole-cell patch clamp technique. After measuring the normal current in A2780-SP cells, 10 µM of TM was applied for 5 min, and the change in the current amplitude by TM was measured using the same voltage.

**Figure 3 cells-10-00918-f003:**
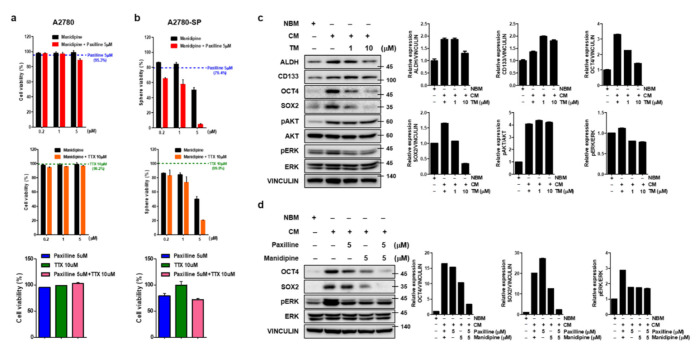
Simultaneous inhibition of BKCa and calcium channel in ovarian CSCs enhances cell death and further reduces stemness. ATP-based cell viability of (**a**) A2780 and (**b**) A2780-SP cells measured when the calcium and BKCa channels were simultaneously inhibited by co-treatment with 0.2, 1, or 5 µM manidipine (calcium channel blocker) and 5 µM of paxilline (BKCa channel inhibitor), and compared to the cell inhibition resulting from treatment with manidipine alone. ATP-based cell viability was measured when the calcium and sodium channels were simultaneously inhibited by co-treatment with 0.2, 1, or 5 µM manidipine (calcium channel blocker), and 10 µM of TTX (sodium channel blocker), and compared to the cell inhibition resulting from treatment with manidipine alone. ATP-based cell viability when the BKCa channel and sodium channels were simultaneously inhibited was measured by cotreatment with 5 µM of paxilline and 10 µM of TTX compared to individual treatments. (**c**) TM was applied to A2780-SP cell CSC culture on CM media for 24 h and cell lysates were collected. The expression level of stemness-related markers (ALDH, CD133, OCT4, and SOX2) and the phosphorylation level of AKT and ERK were analyzed by Western blot. Vinculin was used as a loading control for the cell lysates. (**d**) BKCa and calcium channels were simultaneously inhibited by co-treatment with 5 µM of paxilline and 5 µM of manidipine for 24 h on A2780-SP cell CSC culture with CM media. Cell lysates were collected and the expression levels of OCT4 and SOX2 and the phosphorylation level of ERK were analyzed by Western blot.

**Figure 4 cells-10-00918-f004:**
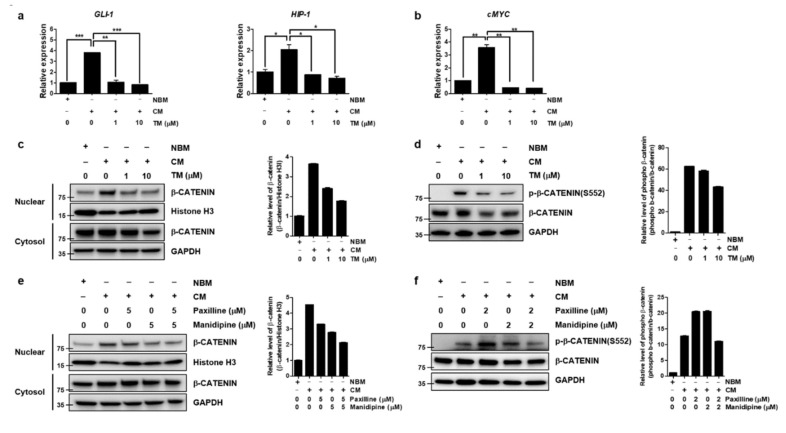
TM further reduces Wnt/β-catenin signaling by simultaneously inhibiting the BKCa and calcium channels in ovarian CSCs. After TM was applied to CSC culture of A2780-SP cells for 24 h, (**a**) relative mRNA expression change of *GLI-1, HIP1* and *PTCH-1* for HH pathway, and (**b**) relative mRNA expression change of *c-Myc* for Notch pathway were analyzed. The data are expressed as the mean ± SD of two or three independent experiments. * *p* < 0.05, ** *p* < 0.01, *** *p* < 0.001. (**c**) TM was applied to CSC culture of A2780-SP cells for 24 h and the reduced translocation of β-catenin into the nucleus by TM was analyzed by Western blot. Histone H3 and GAPDH were used as loading controls for the nuclear and cytosol fractions, respectively. (**d**) Reduced β-catenin phosphorylation at S552 by TM on CSC culture of A2780-SP cells was analyzed by Western blot. GAPDH was used as a loading control for cell lysates. (**e**) The BKCa and calcium channels were simultaneously inhibited by cotreatment with 5 µM of paxilline and 5 µM of manidipine for 24 h in CSC culture of A2780-SP cells, compared to individual treatments. β-catenin translocation into the nucleus was analyzed by Western blot. (**f**) The BKCa and calcium channels were simultaneously inhibited by cotreatment with 2 µM of paxilline and 2 µM of manidipine for 24 h on CSC culture of A2780-SP cells, compared to individual treatments. Reduced β-catenin phosphorylation at S552 by simultaneous inhibition of the BKCa and calcium channels was analyzed by Western blot.

**Figure 5 cells-10-00918-f005:**
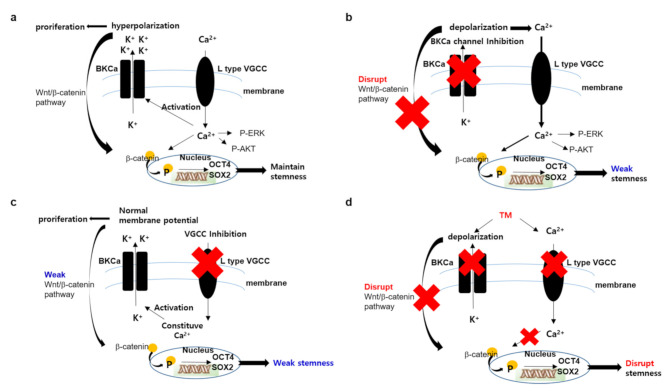
Proposed mechanism of TM in ovarian CSCs. (**a**) In ovarian CSCs environment, the amplified Ca^2+^ channel functions to maintain stemmness and cell proliferation through the AKT, ERK or Wnt pathway. The amplified BKCa channel functions to further activate the wnt/ β-catenin pathway through hyperpolarized membrane potential. (**b**) When the BKCa channel is inhibited by selective inhibitor, such as paxillin, the Ca^2+^ channel is further activated by positive feedback following membrane depolarization. Increasing of intracellular Ca^2+^ concentration is enough to maintain stemness and proliferation. (**c**) When the Ca^2+^ channel is inhibited by a selective inhibitor, such as manidipine, the BKCa channel activity enables the maintenance of a weak stemness and, therefore, requires a stronger Ca^2+^ channel blockade to induce cell death. (**d**) When the Ca^2+^ channel and BKCa channel are blocked simultaneously by TM, though weakly, it is enough to disrupt stemness and induce cell death.

**Table 1 cells-10-00918-t001:** The primers used in these experiments.

Gene Symbol	Forward 5′-3′	Reverse 5′-3′
*GAPDH*	GGAGCCAAAAGGGTCATCAT	GTGATGGCATGGACTGTGGT
*KCNMA1*	TCTTTGCTCTCAGCATCGGTG	CCGCAAGCCGAAGTAGAGAAG
*KCNMB2*	GAGGACCGAGCTATTCTCCTG	TGTTTCCGTGATGGACGCATT
*KCNMB3*	TTCTTGCTCGGAACAACCATT	GATGGCAGTGCAGGTCGATT
*KCNMB4*	GGTCTACGTGAACAACTCTGAG	GGAGGGATATAGGAGCACTTGG
*SCN5A*	TCTCTATGGCAATCCACCCCA	GAGGACATACAAGGCGTTGGT
*SCN8A*	CTCCTGACTGGTCGAAGAATGT	CATGGGTCCCGTAAAAAGGTAA
*SCN1B*	CTGCTGGCCTTAGTGGTCG	GTGAAGGTCTCAGCGTTGGTC
*SCN4B*	CAACAGCAGTGACGCATTCAA	CTCCTTAGTAGAGCCTACCAGAG
*OPRM1*	GCCCTTCCAGAGTGTGAATTAC	GTGCAGAGGGTGAATATGCTG
*OPRK1*	ATCATCACGGCGGTCTACTC	ACTCTGAAAGGGCATGGTTGTA
*CACNA1C*	ATGAACATGCCTCTGAACAGCG	TTCAGAAGCAGCGGACACAGC
*CACNA1D*	TGTGATTTGCAAGATGACGAGCC	TGAGATTGGCATTTGTTGAGGTG
*CACNA1F*	ATTTCTGTGTCTCTGCCTGTCG	TAGGTAGGAAAGCCGATCAGG
*CACNA1G*	TGTCTCCGCACGGTCTGTAA	AAGCCGGTTCCAAGTGTCTC
*CACNA1H*	TTGGGTTCCGGTCGGTTCT	ATGCCCGTAGCCATCTTCA
*CACNA1I*	ATCGGTTATGCTTAGGATTGTCA	TGCTCCCGTTGCTTGGTCTC
*GLI-1*	AGCGTGAGCCTGAATCTGTG	CAGCATGTACTGGGCTTTGAA
*HIP-1*	ACACGCCAGAACGTGCATA	CACTGCGTTGCTAGACAGAG
*c-MYC*	GGCTCCTGGCAAAAGGTCA	CTGCGTAGTTGTGCTGATGT

## Data Availability

Please refer to suggested Data Availability Statements in section “MDPI Research Data Policies” at https://www.mdpi.com/ethics.

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
