# Peer review of "Repositioning Trimebutine Maleate as a Cancer Treatment Targeting Ovarian Cancer Stem Cells"

_cells, 2021, doi:10.3390/cells10040918_

Round 1

Reviewer 1 Report

The manuscript by Lee and colleagues discusses the potential of repositioning trimebutine maleate as a cancer targeting method. The authors have conducted several experiments supporting their hypothesis that TM is effective in inhibiting CSC proliferation and progression in OVC. However, it seems the manuscript can be improved by providing some additional explanations and experiments.

Major Comments:

  1. The authors should show the difference in cell morphology of A2780 and A2780-SP cells (as well as the other cell lines used in this manuscript). This figure is necessary to show the sphere-forming ability of A2780-SP cells.
  2. The authors should add western blot (protein expression level) images for figures 2a and 2b.
  3. In Figure 3a and b, the experiment of single-agent treatment (paxilline or TTX) needs to compare with the results of combined treatments.
  4. Based on the sup S1c, the viability of A2780-SP cells and other CSCs treated with TM showed quite different effects. Thus, the authors suggest that TM shows a stronger effect on subpopulations expressing other traits or markers of ovarian CSCs. They should describe the reason in the discussion or need to perform some experiments.
  5. The authors need to confirm the changes in protein levels for GLI-1, HIP-1, and cMYC in figures 4 and 4b.
  6. The authors state that inhibition of BKCa channels and Ca channels by paxilline and manidipine together has a stronger cell inhibitory activity than TM alone. How is this an efficient method of suppressing ovarian CSC’s stemness and proliferation?
  7. Also, the authors mention that Na channels are inhibited by using TM. Thus, how is using or repositioning TM to suppress cancer stemness and proliferation a selective method? What is the advantage of repositioning TM to targeting ovarian cancer stem cells over synergistically using paxilline and manidipine?
  8. Has TM been used in other cancers as well? Why did the authors specifically choose ovarian cancer as the subject of this study? Although TM is widely used to treat IBS and the related bowel syndromes, there is a lack of connection between TM and ovarian cancer in the manuscript. The authors should make clear connections between TM and ovarian cancer in the introduction section.
  9. The authors should describe in detail or perform additional experiments to show the direct relationship between BKCa or Ca channels and the signaling component of the signaling pathway linking the reduction of the stemness-related transcription factor.

Minor Comments:

  1. The authors should go over the manuscript thoroughly and revise the overall English of the paper.
  • Typo: line 263, line 375
  • Clarify the sentence: lines 275-277
  • Punctuations throughout the paper.
  • All Ca2+, Na+, and other ions should be written in their correct form throughout the manuscript.

Supplementary Figure 2 lacks figure legend.

Reviewer 2 Report

The authors address one of the main unmet need in cancer field and suggest novel therapeutic options for ovarian cancer bearing woman. They exploit tumor cell metabolism, with particular attention to cancer stem cell subset responsible for current anti-tumor therapies refractoriness and tumor relapse.

The paper is well organized and described.

1) One of the major concern of this study is that the culture condition of A278-SP could influence the effects of TM treatment and the description of the mechanism exploited by it. Indeed, the absence of serum and adhesion could modify cell metabolism and channel functionality. The authors should performed the main experiments using the cell line maintained in normal culture condition adding a staining for CD44/c-kit, whose co-expression has been demonstrated reliable marker for CSC identification in ovarian cancer (see Bapat Z or Pastò A publication). In this manner, by flow cytometry approach it is possible to restrict the analysis to the CSC subset. In addition, when possible, the cells could be FACS sorted after treatment (for instance to performed WB analysis).

In particular, the authors should repeat the simultaneous inhibition of the different channels in the A2780 cells and analyze the cell viability by flow cytometry with a specific dye (i.e. Live/Dead or similar) in CD44+/c-kitpos and CD44+/c-kitneg cells.

2) The authors conclude that TM action is mediated by WNt/B catenin pathway. However, they just showed a reduction in the nuclear accumulation of B catenin and its phosprilation after treatment. These results are not enough to prove the regulatory mechanism.

Minor revision:

Line 79 a dot is missing

Line 263, They should be not in capital letter

Add marker to the WB in order to appreciate the right height of the bands

Round 2

Reviewer 1 Report

  1. The authors have responded well to the question raised.
  2. The authors have responded to the question raised. However, WB data would further strengthen your conclusion.
  3. The authors need to add following graphs to figure 3a and figure 3b for clear comparison. Although Paxxiline, manidipine, and TTX alone in A2780 and A2780-SP did not lead to significant reduction in cell viability (as shown in the last panel of figure 3a and 3b), because cell viability in percentages is a relative representation of data, all combination sets must include single treatment for each drugs to solidly state synergism between the two drugs.
    • First panel of figure 3a and figure 3b: cell viability of single treatment of paxilline.
    • Second panel of figure 3a and figure 3b: cell viability of TTX 10 uM.
  4. The authors should unify the manuscript’s format in referring to supplementary figures (Line 526): “TM shows a stronger effect on A2780-SP cells than other ovarian CSCs based on the sup S1c.”
  5. The authors have responded to the question raised. However, the author should add WB data to further strengthen your conclusion.
  6. The authors need to revise their discussion part in lines 499-502 for clarity. The authors show that TM has less ability to inhibit calcium channels compared to manidipine and other CCB drugs (lines 497-499). Also, they state that TM’s weaker potency may be the result that contributes to the ‘small effect on cell growth inhibition compared to CCB drugs’ (lines 499-500). However, the authors again state that there are other factors that might contribute to the ‘superior anti-CSC effects of TM’ (line 501-502). There is confusion in what the authors are trying to deliver to the readers. The authors should revise this discussion part.
  7. The authors have responded well to the question raised. The authors should rephrase the answer and add it to the discussion section of the manuscript to strengthen their thesis.
  8. The authors have responded well to the question raised.
  9. The authors have responded well to the question raised.

Reviewer 2 Report

The Authors addressed all the revisions asked.

As expected the TM effect on the FACS isolated CSC subset is less potent than on the SP population, probably because the different culture condition. The authors could try to maintain CSCs, FACS sorted according to the markers used, in the same culture condition as SP and evaluate the effects of TM in such culture condition.

This would implement their results.
